# Long-Term Exposure to Supraphysiological Levels of Testosterone Impacts Rat Submandibular Gland Proteome

**DOI:** 10.3390/ijms25010550

**Published:** 2023-12-31

**Authors:** João Valente-Santos, Rui Vitorino, Cláudia Sousa-Mendes, Paula Oliveira, Bruno Colaço, Ana I. Faustino-Rocha, Maria João Neuparth, Adelino Leite-Moreira, José Alberto Duarte, Rita Ferreira, Francisco Amado

**Affiliations:** 1LAQV-REQUIMTE, Department of Chemistry, University of Aveiro, 3810-193 Aveiro, Portugal; jmvsantos@ua.pt (J.V.-S.); ritaferreira@ua.pt (R.F.); 2Department of Medical Sciences, Institute of Biomedicine-iBiMED, University of Aveiro, 3810-193 Aveiro, Portugal; rvitorino@ua.pt; 3UnIC@RISE, Department of Surgery and Physiology, Faculty of Medicine of the University of Porto, Alameda Prof. Hernâni Monteiro, 4200-319 Porto, Portugal; ccsousam@gmail.com (C.S.-M.); amoreira@med.up.pt (A.L.-M.); 4Centre for Research and Technology of Agro Environmental and Biological Sciences (CITAB), Inov4Agro, University of Trás-os-Montes and Alto Douro (UTAD), Quinta de Prados, 5000-801 Vila Real, Portugal; pamo@utad.pt (P.O.); anafaustino.faustino@sapo.pt (A.I.F.-R.); 5Veterinary and Animal Research Centre (CECAV), Associate Laboratory for Animal and Veterinary Sciences (AL4AnimalS), Department of Animal Science, University of Trás-os-Montes and Alto Douro (UTAD), Quinta de Prados, 5000-801 Vila Real, Portugal; bcolaco@utad.pt; 6Comprehensive Health Research Center, Department of Zootechnics, School of Sciences and Technology, University of Évora, 7006-554 Évora, Portugal; 7Laboratory for Integrative and Translational Research in Population Health (ITR), Research Center in Physical Activity, Health and Leisure (CIAFEL), Faculty of Sports, University of Porto, 4200-450 Porto, Portugal; mjoao.neuparth@ipsn.cespu.pt (M.J.N.); jarduarte@fade.up.pt (J.A.D.); 8TOXRUN—Toxicology Research Unit, University Institute of Health Sciences, CESPU, CRL, 4585-116 Gandra, Portugal

**Keywords:** salivary glands, submandibular gland’s proteome, testosterone, rats, GeLC-MS/MS, kallikrein-3

## Abstract

The salivary glands play a central role in the secretion of saliva, whose composition and volume affect oral and overall health. A lesser-explored dimension encompasses the possible changes in salivary gland proteomes in response to fluctuations in sex hormone levels. This study aimed to examine the effects of chronic exposure to testosterone on salivary gland remodeling, particularly focusing on proteomic adaptations. Therefore, male Wistar rats were implanted with subcutaneous testosterone-releasing devices at 14 weeks of age. Their submandibular glands were histologically and molecularly analyzed 47 weeks later. The results underscored a significant increase in gland mass after testosterone exposure, further supported by histologic evidence of granular duct enlargement. Despite increased circulating sex hormones, there was no detectable shift in the tissue levels of estrogen alpha and androgen receptors. GeLC-MS/MS and subsequent bioinformatics identified 308 proteins in the submandibular glands, 12 of which were modulated by testosterone. Of note was the pronounced upregulation of Klk3 and the downregulation of Klk6 and Klk7 after testosterone exposure. Protein–protein interaction analysis with the androgen receptor suggests that Klk3 is a potential target of androgenic signaling, paralleling previous findings in the prostate. This exploratory analysis sheds light on the response of salivary glands to testosterone exposure, providing proteome-level insights into the associated weight and histological changes.

## 1. Introduction

Salivary gland remodeling can be strongly influenced by androgens, which can have profound effects on both the structure and function of these glands, ultimately impacting saliva production and altering its composition [1,2,3]. Androgens exert their effects by binding to androgen receptor (AR), triggering a series of biochemical reactions that modulate gene expression and bring about diverse physiological changes [4,5]. The presence of AR in salivary glands suggests that androgens can directly influence the activity of these glands. Studies have demonstrated that androgens can enhance the proliferation and differentiation of salivary gland cells, resulting in the enlargement of the glands and in an increase in their overall volume. In fact, testosterone, in particular, exerts effects on the salivary glands, influencing glandular size and function, as well as modulating saliva synthesis, secretion, and composition [2,6,7,8]. This can, at least in part, account for the observed sexual dimorphism in the submandibular glands of rodents, with mice being a notable example [9,10]. Nevertheless, the exact molecular mechanisms underlying the impact of androgens on salivary gland remodeling are still being investigated, and further studies are necessary to establish a comprehensive understanding of the role played by androgens in this complex process.

Testosterone has gained significant attention for its application in several settings, such as its anabolic role in sports enhancement, its use in hypogonadism treatment, and its application in the management of castration-resistant prostate cancer [11,12]. In fact, testosterone has been increasingly prescribed to middle-aged and older men diagnosed with low circulating testosterone levels [13]. Despite its wide applications, concerns have emerged regarding potential risks associated with testosterone therapy, particularly an increased risk of cardiovascular diseases. Consequently, a comprehensive understanding of the effects of testosterone treatment on various aspects of health is imperative to ensure informed prescription practices [14,15,16,17]. One area of interest is the influence of testosterone on the structure and functionality of salivary glands given its effects in modulating their size and function. In the natural aging process, the levels of sex hormones, including testosterone, gradually decline, which can contribute to age-related dysfunction of the salivary glands, leading to conditions such as xerostomia (dry mouth) and impaired saliva production [1,18,19,20,21,22,23,24,25]. Given that the salivary glands play a crucial role in maintaining oral health and overall well-being, exploring the impact of testosterone therapy on the salivary glands can provide valuable insights into potential therapeutic approaches to managing salivary gland-related disorders in older individuals.

Therefore, the aim of our study was to investigate how chronic exposure to supraphysiological testosterone levels affects salivary gland remodeling, with a particular focus on proteome adaptation. For this purpose, male Wistar rats were implanted subcutaneously with devices that release testosterone for 47 weeks, which corresponds to approximately 30 years in human age [26]. At the end, their submandibular glands were harvested for histological and molecular characterization. SDS-PAGE-LC-MS/MS followed by bioinformatics analysis was performed to explore the proteome adaptations promoted by testosterone on these glands. The submandibular glands were chosen due to their role in regulating the majority of total unstimulated saliva (≈65%) [27,28], and since they have been previously shown to be highly responsive to sex hormones, particularly testosterone [6].

## 2. Results

### 2.1. Impact of Testosterone Exposure on Anthropometric and Systemic Biochemical Parameters

Forty-seven weeks of exposure to exogenous testosterone (TEST group) caused a significant reduction in body weight (BW) when compared with age-match CTRL animals (*p* < 0.01; Figure 1A). When examining the salivary glands, no changes in parotid mass were noticed, whereas a significant increase in submandibular gland weight was observed in TEST group (*p* < 0.05; Figure 1B). Moreover, the submandibular gland mass-to-body weight ratio was higher in TEST group when compared with the CTRL one (*p* < 0.01; Figure 1C), which suggests tissue hypertrophy and/or hyperplasia.

The levels of testosterone were measured in serum samples, and the data confirmed that animals exposed for 47 weeks to exogenous testosterone exhibited approximately 15 times higher levels of testosterone in their circulation compared to the age-match control animals (*p* < 0.0001; Figure 2A). A strong positive correlation between submandibular gland mass-to-body weight ratio and testosterone levels was observed (*p* = 0.006; Appendix A). The levels of 17beta-estradiol were also measured in these animals, revealing significantly elevated levels (30 times higher) in the TEST rats. These findings provide strong evidence for the aromatization of testosterone into 17beta-estradiol, a reaction primarily occurring in adipose tissues [29].

To gain further insights into the long-term systemic adaptations resulting from these supraphysiological levels of sex hormones, the biochemical markers of metabolism and inflammation were assessed (Figure 2). In fact, the TEST rats presented no significant alterations in serum levels of the negative acute-phase protein albumin compared with the CTRL group (*p* > 0.05; Figure 2B). Nevertheless, testosterone exposure led to a significant reduction in serum total protein levels (*p* < 0.05; Figure 2C). Regarding glucose levels, no significant differences were observed between the two groups (*p* > 0.05; Figure 2D). Likewise, cholesterol levels showed no significant differences between the groups (*p* > 0.05; Figure 2E). Lastly, testosterone administration caused a significant reduction in the circulating levels of triglycerides in the TEST rats (*p* < 0.001; Figure 2F).

### 2.2. Effect of Testosterone Exposure on Submandibular Gland Histology

Representative histological sections of submandibular glands obtained from the CTRL and TEST rats and stained with H&E are presented in Figure 3A. Chronic exposure to testosterone induced histological changes in the submandibular glands, primarily marked by the expansion of granular ducts with more eosinophilic secretory granules. Indeed, the quantitative analysis of the ductal-to-total tissue area evidenced a significant increase following testosterone exposure (*p* < 0.05; Figure 3B). Granular ducts are testosterone-sensitive, which may account for the morphologic variations in this gland [9]. These adaptations may have resulted in an overall enlargement of the glandular tissue, which may explain, at least in part, the increase in the submandibular gland-to-body weight ratio observed in the TEST rats.

### 2.3. Impact of Testosterone Exposure on the Levels of Androgen Receptor and Estrogen Receptor Alpha in Submandibular Glands

To assess the potential impact of elevated circulating levels of sex hormones on the histological and molecular remodeling of submandibular glands, the expression of ARs and ERα was assessed through Western blotting on whole tissue extracts. The data revealed no significant differences in the levels of both receptors between the CTRL and TEST groups (*p* > 0.05; Figure 4). Thus, our findings suggest that there is no direct involvement of the testosterone and 17beta-estradiol signaling pathways in the remodeling of submandibular glands in rats chronically exposed to exogenous testosterone. This lack of differences in AR and ERα expression may indicate potential resistance of submandibular glands to the effects of exogenous testosterone. Nevertheless, there were also no differences in AR expression reported in the submandibular glands of castrated rats [30].

### 2.4. Proteomic Characterization of Submandibular Glands

SDS-PAGE-LC-MS/MS allowed for the identification of 308 distinct proteins with at least two peptides and a false discovery rate (FDR) of 1% (Appendix A), most of which belong to the biological processes’ chaperone-mediated protein transport (GO:0061741), calcium ion transport (GO:1903515), metabolism (GO:0008152), and response to stimulus (GO:0050896), according to STRING (https://string-db.org/, accessed on 5 June 2023). The majority of the identified proteins are located in the mitochondria and endoplasmic reticulum. Of the 308 identified proteins, 65% are present in both groups of animals, 97 proteins are exclusive to the CTRL group (i.e., present in at least two of the six CTRL rats and not in TEST rats), and 12 proteins are exclusive of TEST group (Appendix A).

The MS data were further analyzed with MetaboAnalyst (https://www.metaboanalyst.ca/, accessed on 9 May 2023) to visualize group separation through both unsupervised PCA and supervised PLS-DA. The PCA analysis showed no clustering of proteome data among the groups (Figure 5A) whereas PLS-DA indicated mild separation between the CTRL and TEST groups (Figure 5B), suggesting the presence of distinct protein expression patterns that may differentiate these two groups.

To identify the most abundant proteins in each group and gain insights into the biological processes associated with them, a quantitative analysis was conducted based on the Log2 ratio of iBAQ values between the CTRL and the TEST rats. Only proteins present in at least four animals in each group were considered in this analysis, which translates to a total of 87 proteins analyzed (Figure 6).

From this analysis, 26 proteins were found to be downregulated and 14 were upregulated, while the remaining 47 proteins had no significant expression differences between groups. The top 10 most abundant proteins in the submandibular glands of the TEST rats were Klk3, Prol1, Ldha, Acta2, Atp5b, Actb, Atp1a1, Atp2a3, Hibadh, and Hsp90ab1. To provide further context and understanding of the functions of these proteins, their corresponding biological processes were annotated using information obtained from Perseus, UniProt, and Gene Ontology (Appendix A). Overall, these proteins seem to be mostly involved in metabolism, ion transport, proteolysis, protein folding, and cell adhesion. From these proteins, Klk3, also known as prostate serum antigen (PSA) in humans, stands out as the protein more susceptible in the submandibular glands to testosterone exposure, with a Log2 fold change of a two-fold increase (meaning approximately a four-fold ratio elevation, as shown in Table 1) compared to the CTRL rats (Figure 6). In castrated rats, no changes in the expression of Klk3 and other Klks were observed [30]. According to *UniProt*, this Ser endopeptidase cleaves Met-Lys and Arg-Ser bonds in kininogen to release Lys-bradykinin. Still, other substrates are recognized by this enzyme, including protease-associated receptors and growth factors [31].

In order to better understand the effect of testosterone exposure on submandibular proteome remodeling, a PPI network centered on ARs and ERα (and considering the unique and upregulated proteins found in TEST group) was performed with the *STRING* server (Figure 7).

Regarding ERα, or ERS1, Figure 7 highlights a stronger direct interaction with Hsp90ab1 (a molecular chaperone that promotes the maturation and proper regulation of specific target proteins involved, for instance, in cell cycle control and signal transduction), Actb (which is involved in various types of cell motility and is ubiquitously expressed in all eukaryotic cells), and Acta2 (an actin involved in cell motility and differentiation). Concerning ARs, a stronger connection is again visible with Hsp90ab1, Actb, and Klk1c3, which also stands for Klk3 (the glandular kallikrein identified in a higher amount in the MS identification).

To gain insights on the role of kallikreins in the proteome remodeling of submandibular glands, zymography analysis was performed. This electrophoretic methodology allows for the visualization of protease activity in a gel matrix containing a protein source such as collagen. The data showed a similar proteolytic profile between the CTRL and TEST groups, with a prominent band with proteolytic activity observed in the molecular weight range of 25–40 kDa (Appendix A). Considering the information from the MEROPS database (ebi.ac.uk/merops), and the MS-based proteome data obtained in the present study (Appendix A), it is plausible to speculate that the proteolytic activity seen in the zymography gel could be attributed, at least in part, to kallikreins. Kallikreins are a group of serine proteases known to be involved in various physiological processes, including tissue remodeling and inflammation [32,33,34]. Table 1 provides a description of the kallikreins identified by SDS-PAGE-LC-MS/MS in the present study, their corresponding molecular weights (resorting to UniProt), and their fold changes in TEST/CTRL. According to the information retrieved from Perseus Software (v2.0.3.1), the main biological processes associated with these kallikreins include positive regulation of the acute inflammatory response, the regulation of cell proliferation, and the regulation of vasoconstriction. Further analysis using STRING and UniProt revealed that these kallikreins are mainly located in the secretory granule (GO:0030141), which aligns with their function as secreted enzymes.

The data highlight the unique upregulation of Klk3 and the downregulation of Klk6 and Klk7. Based on data obtained from the Perseus, UniProt, and Gene Ontology resources, Klk3 primarily participates in the androgen receptor signaling pathway, proteolysis, and the negative regulation of angiogenesis. This suggests that Klk3 may play a role in the regulation of hormonal signaling and tissue remodeling. On the other hand, Klk6 and Klk7 are involved in various biological processes, including the regulation of systemic arterial blood pressure, immune and inflammatory responses, zymogen activation, tissue homeostasis, proteolysis, protein binding and processing, the regulation of serine-type endopeptidase, and the positive regulation of antibacterial peptide production. A negative correlation was identified between circulating testosterone levels and the content of Klk7 in the submandibular glands (*p* < 0.05; Appendix A). This underscores the inhibitory impact of this sex hormone on the expression of Klk7.

## 3. Discussion

Testosterone supplementation occurs in various settings, including sports and specialized clinics. One potential scenario where intentionally elevating testosterone levels may be necessary is in medical treatments such as testosterone replacement therapy in older men, in men with prostate cancer, or in cases of hypogonadism [35,36,37,38]. However, the potential side effects of its supplementation are poorly explored, particularly regarding their impact on the remodeling of salivary glands and, consequently, on oral health. The present study aimed to shed light on the impact of long-term exposure to supraphysiological levels of testosterone on the remodeling of submandibular glands. In rodents, the testosterone sensitivity of these glands is a key factor contributing to the observed sexual dimorphism noticed at the histological levels and in terms of secretory activity [9]. The utilization of a rat model was deemed necessary due to ethical limitations associated with obtaining salivary gland samples from humans exclusively exposed to supraphysiological levels of testosterone without concurrent conditions. Although rats possess serous glands, in contrast to the mixed glands found in humans, they have been extensively employed in the study of salivary gland pathophysiology and hormonal modulation. This choice is grounded in the fact that rat salivary glands are subject to multihormonal regulation, rendering them a well-established and pertinent model for investigating these physiological processes [9].

In fact, histological examination revealed signs of a larger ductal area, stemming from hypertrophy of granular ducts, thereby leading to overall enlargement of the glandular tissue (Figure 3). Indeed, the heightened testosterone sensitivity observed in these ducts, as opposed to other components of the parenchyma in the submandibular gland, may underlie the reported sexual dimorphism in rodents. Larger granular ducts lined with a taller epithelium, and more eosinophilic secretory granules, were observed in males compared to their female counterparts [39]. These histological adaptations may explain, at least in part, the observed increase in the submandibular gland-to-body weight ratio in the TEST rats. Interestingly, despite no changes in the tissue levels of androgen and estrogen alpha receptors (Figure 4), the proteome of the submandibular glands was modulated by exogenous testosterone. This modulation involved the upregulation of proteins associated with metabolism, ion transport, proteolysis, protein folding, and cell adhesion in the submandibular glands of the TEST rats. Among these proteins, Klk3 stands out due to its elevated levels (four-fold) following exposure to testosterone (Table 1), and its regulation by AR signaling (Figure 7). To the best of our knowledge, this study represents the first identification of Klk3 as a potential target influenced by the testosterone-mediated remodeling of submandibular glands. The testosterone-induced modulation of the submandibular gland’s proteome may reflect the enhancement in the secretory activity of granular ducts (Figure 3).

In contrast to the high overexpression of Klk3, Klk6 and Klk7 were found to be downregulated in the submandibular glands following testosterone exposure (Table 1). Klk7 is predominantly expressed in the skin and mainly regulates the development of adipocytes, playing an important role in adipogenesis and lipid metabolism in adipose tissue [40,41]. Moreover, Klk7 overexpression has been documented in various malignant tumors, including ovarian cancer, pancreatic cancer, thyroid cancer, and colon cancer. Conversely, in prostate cancer and breast cancer, the expression of Klk7 is downregulated [42]. The significant negative correlation observed between Klk7 levels in the submandibular glands and testosterone levels (Appendix A) implies that testosterone signaling plays a suppressive role in Klk7 expression within the submandibular glands. To our knowledge, such a regulatory mechanism has not been investigated previously. Klk6 is involved in activating proliferation, extracellular matrix remodeling, and tumor invasion, while also promoting the growth and invasion of gastric cancer cells [43,44], as well as ovarian and colorectal cancer [45]. In salivary gland tumors, this kallikrein was found to be downregulated [45]. Unlike Klk3, Klk6 was reported to be more responsive to estrogens [46]. This may explain, at least in part, the downregulated expression of Klk6 in the submandibular glands after testosterone exposure (Table 1), showing its potential as a possible biomarker for salivary gland-related diseases.

Kallikreins constitute a family of serine proteases that are distributed in various tissues and implicated in several pathological disorders, including cancer and neurological disorders. In fact, kallikrein genes represent the largest group of proteases within the human genome. Specifically, kallikreins are abundant in the salivary glands since they are involved in proteolysis, proteolytic cascades, and regulation of the acute inflammatory response, which affects saliva composition and, thus, oral health [32,33,34,47,48,49]. In rats, their production primarily takes place in the granular ducts, and they can be secreted from both basolateral and apical surfaces, contributing significantly to the overall saliva composition [50]. In fact, our data emphasized the contribution of this family of proteases to the proteolytic activity of these glands, as noticed in the zymographic gel profile (Appendix A). Nevertheless, exposure to testosterone did not appear to modulate the proteolytic activity assessed by zymography, possibly due to the simultaneous upregulation of some kallikreins and downregulation of others which have similar molecular weights (Table 1).

Klk3 is a well-known target of AR, primarily recognized in prostate tissue [46,51]. However, its presence and significance in submandibular glands have been relatively unexplored. In fact, salivary glands are expected to have low protein levels of this kallikrein (approximately 33.9 ppm), according to the PAXdb database (Appendix A). Moreover, the expression of human Klk3 was not detected in normal salivary gland tissue nor in salivary gland tumors, using immunohistochemistry assays [52]. However, using in situ hybridization, Klk3-encoding transcripts were detected in the serous epithelial cells of submandibular glands [53]. The finding of Klk3 in the salivary glands, along with its upregulation following chronic exposure to testosterone, may have varying implications depending on the context.

In prostate glands, Klk3 has been extensively investigated. Similar to the observations in submandibular glands (Figure 1), elevated levels of Klk3 have been associated with an increase in prostate gland weight due to hyperplasia [54,55]. When testosterone stimulates the growth of the prostate glands, it may produce more Klk3 as it enlarges, potentially leading to higher Klk3 levels [31,56]. In the case of prostate hyperplasia, the activation of ARs in prostatic cells triggers growth, resulting in the transcription of specific target genes and the production and secretion of peptide growth factors [54,57].

Long-term exposure to testosterone can indeed result in increased levels of Klk3, primarily mediated through the androgen signaling pathway. This pathway is pivotal in various tissues influenced by hormones, such as the prostate, where multiple molecular mechanisms involving AR and ERs are at play. Similar to the salivary glands, AR signaling significantly contributes to the development, growth, and function of the prostate [12,58]. Upon binding to androgens, ARs translocate into the nucleus and activate gene expression, leading to the synthesis of specific proteins that promote cell proliferation and differentiation and the maintenance of prostate tissue [3,4,59,60]. Estrogen can either be locally synthesized within the prostate or derived from the systemic circulation. When estrogen binds to ERs, it triggers signaling pathways that affect proliferation, apoptosis, and inflammation [61,62,63]. The interaction between the AR and ER signaling pathways is complex, and they can impact each other’s activity. For instance, the balance between androgen and estrogen levels can impact the development and progression of local diseases [64]. In the present study, the absence of signs of inflammation (Figure 2) and cellular injury or death (Figure 3) suggests that AR and ER signaling may be primarily involved in cell proliferation, leading to the mass gain observed in the salivary glands (Figure 1).

Within the nucleus, the testosterone-AR complex regulates gene transcription, resulting in the synthesis of proteins, including Klk3 in prostate cells [3,59,60,65]. A similar mechanism may be at play in the submandibular glands. However, no changes in the levels of sex hormones receptors, including ARs, were observed after 47 weeks of testosterone exposure (Figure 4). Nevertheless, prolonged elevation of testosterone levels may lead to the development of testosterone resistance, where the hormone’s effects become diminished. Testosterone resistance can be attributed to several mechanisms, including reduced cellular responsiveness to the hormone or changes in the expressions of co-regulators and other proteins involved in AR signaling [65,66]. AR gene activation is often associated with complex regulatory modifications and alterations, including gene amplification, mutations, and isoforms formation, which have been linked, for instance, to hormonal resistance [67,68,69,70,71]. However, one limitation is that our analysis focused on whole tissue extracts rather than specific subcellular compartments, particularly the nucleus, where the complex hormone/hormone receptor regulates gene expression. Thus, future studies should employ experimental approaches that allow for a more targeted examination of testosterone-AR activity. Further exploration of sex hormone signaling and its eventual resistance mechanisms is warranted in future studies to gain a deeper understanding of its contribution to submandibular gland remodeling and, consequently, its impact on overall glandular health outcomes.

Overall, these results underscore the susceptibility of the submandibular glands to prolonged exposure to testosterone. This is evidenced by an increased ratio of submandibular gland mass to body weight due, at least in part, to the enlargement of granular ducts, and associated with an increased abundance of proteins central to metabolism, immune response, and ion transport. Of note, Klk3 emerges as a transcriptional target of AR in the submandibular glands, potentially opening up molecular pathways reminiscent of those observed in the prostate glands. The marked prominence of Klk3 in the mandibular glands after 47 weeks of testosterone exposure likely implies long-term tissue adaptation to elevated hormone levels, even in the absence of detectable shifts in AR content. This raises the need to explore the dynamics of testosterone, and AR and Klk3 signaling, in more detail in future research. Further exploration of the inhibitory effect of testosterone on Klk6 and Klk7 expression, and their potential impact on susceptibility to salivary gland diseases, also warrants careful consideration in future studies. A focused investigation of this signaling pathway will undoubtedly enrich our understanding of how external sources of testosterone influence restructuring and pave the way for predicting the effects of testosterone-centered treatments on oral well-being. These insights hold promise for informing clinical strategies and interventions aimed at optimizing oral health in individuals undergoing testosterone-based therapies.

## 4. Materials and Methods

### 4.1. Animal Experimentation

All the animal experiments conducted herein were approved by the Institutional Animals Ethics Committee and by the Portuguese national authorities (Direção Geral de Alimentação e Veterinária, approval number 021326).

Four-week-old male Wistar Unilever rats (*Rattus norvegicus*; n = 24) were purchased from Charles River Laboratories (France), acclimatized for one week prior to the start of the experiment, and kept at the animal facilities of University of Trás-os-Montes and Alto Douro, under the following controlled environmental conditions: temperature (22 ± 2 °C) and humidity (50 ± 10%). A 12h light (20.00–08.00 h) and dark cycle was maintained throughout the study. Animals were allowed to access food and water ad *libitum* (Mucedola 4RF21^®^, Milan, Italy).

Animals were randomly divided into two groups as follows: control group (CTRL, n = 10) and testosterone group (TEST, n = 14). At 12 weeks of age, animals from the TEST group received subcutaneous administration of the anti-androgenic drug flutamide (50 mg/Kg; TCI Chemicals^®^, Portland, OR, USA) for 21 consecutive days. Twenty-four hours after the last flutamide administration, testosterone propionate (TCI Chemicals^®^, Portland, OR, USA) was dissolved in corn oil and subcutaneously administered to the animals at a dose of 100 mg/kg. Forty-eight hours later, they were intraperitoneally injected with the carcinogen agent N-Methyl-N-nitrosourea (MNU) (Isopac^®^, Sigma Chemical Co., Madrid, Spain) at a dose of 30 mg/kg. Two weeks later, silastic tubes were filled with crystalline testosterone (Sigma^®^ Chemical Co., Madrid, Spain) and subcutaneously implanted in the interscapular region of animals previously anesthetized with ketamine (75 mg/kg, Imalgene^®^ 1000, Merial S.A.S., Lyon, France) and xylazine (10 mg/kg, Rompun^®^ 2%, Bayer Healthcare S.A., Kiel, Germany) until the end of experimental protocol.

The animals were sacrificed at 61 weeks of age through an intraperitoneal injection of ketamine (75 mg/kg, Imalgene^®^ 1000, Merial S.A.S., Lyon, France) and xylazine (10 mg/kg, Rompun^®^ 2%, Bayer Healthcare S.A., Kiel, Germany), followed by exsanguination via cardiac puncture. The blood samples were allowed to clot and centrifuged at 3000× *g* for 15 min (4 °C). The serum was separated and frozen at −80 °C until use. A complete necropsy was performed, submandibular glands were collected and weighed, and one of the glands was immediately processed for histological analysis and the other was stored at −80 °C for biochemical analysis. No macroscopic, histological, or biochemical signs of lesions in the salivary glands and in other organs (e.g., liver, heart, and kidney) or inflammation were seen in the TEST animals. This animal protocol was previously implemented to study the effect of testosterone on bone microstructure [72].

### 4.2. Biochemical Analysis in Serum Samples

Total protein, albumin, cholesterol, glucose, and triglyceride levels were measured in serum samples using an AutoAnalyzer (PRESTIGE 35i, Cormay PZ, Diamond Diagnostics, Lomianki, Poland). Testosterone concentration was assessed using an ELISA Kit (582701; Caymann Chemical^®^, Michigan, USA), following the manufacturer’s instructions.

### 4.3. Histological Analysis of Submandibular Glands

The submandibular glands were fixed in 4% paraformaldehyde and embedded in paraffin to prepare paraffin blocks, which were sectioned (3 µm) using a manual microtome. After deparaffinization with xylol and hydration with decreasing concentrations of alcohol (100%, 95%, and 75%), slides from each group were stained with hematoxylin and eosin (H&E). Digital images of submandibular gland sections were captured using a ZEISS Axio Scope A1 optical microscope. Morphological changes in the submandibular glands were qualitatively evaluated through a visual inspection of the stained photomicrograph images captured at an objective magnification of ×10. ImageJ Software (v1.54) from NIH was used to quantitatively determine the percentual contribution of ducts to the whole tissue area.

### 4.4. Preparation of Submandibular Gland Extracts

A portion of submandibular gland was homogenized in urea/thiourea buffer (7 M urea, 2 M thiourea, and 0.1% Triton X-100) in a proportion of 50 mg tissue/mL, using a Teflon pestle on a motor-driven Potter-Elvehjem glass homogenizer. The protein content was assayed in the submandibular gland extracts using a commercial RC-DC^TM^ Protein Assay kit (Bio-Rad^®^, Hercules, CA, USA), according to the manufacturer’s instructions and using bovine serum albumin (BSA) as the protein standard.

### 4.5. SDS-PAGE-LC-MS/MS Analysis of Submandibular Gland Extracts

Equal amounts of protein (50 μg) from the submandibular gland extract of each animal were dissolved (1:2) in reduction buffer (0.5 M Tris-HCl pH 6.8, 4% (*w*/*v*) SDS, 15% (*v*/*v*) glycerol, 0.04% (*w*/*v*) bromophenol blue, and 20% (*v*/*v*) β-mercaptoethanol) and incubated at 100 °C for 5 min. Then, the samples were loaded into 12.5% SDS-PAGE gel prepared according to Laemmli [73]. After staining the gels with Coomassie blue, complete lanes were excised from the gel and cut into seven sections for in-gel digestion. After separation, the gel sections were incubated with 25 mM ammonium bicarbonate (AMBIC)/50% acetonitrile (ACN) for 30 min; this procedure was performed twice. Subsequently, 5 mM dithiothreitol (DTT) was added to the gel sections to break the cysteine bonds, and after 45 min at 56 °C, 55 mM iodoacetamide was added to alkylate to block the cysteine residues. After 30 min at room temperature in the dark, the gel portions were washed with 25 mM AMBIC/50% ACN. Gel sections were then dried in a SpeedVac (Thermo Scientific, Waltham, MA, USA). Enzymatic digestion was performed with trypsin (90057; PierceTM Trypsin Protease MS-grade) overnight at 37 °C. The resulting peptides were extracted with 2% formic acid/50% ACN. This procedure was repeated three times, and the peptides were dried in a SpeedVac. The dried peptides were resuspended in 40 μL of 2% formic acid and centrifuged at 15,000× *g* for 10 min. A total of 2 μL of the supernatant was injected into a PepMap C18 column.

Protein identification was performed using a nanoLC-MS/MS, an instrument consisting of an Ultimate 3000 liquid chromatography system coupled with a Q-Exactive Hybrid Quadrupole-Orbitrap mass spectrometer (Thermo Scientific). Approximately 500 ng of peptides were loaded onto a trap cartridge (Acclaim PepMap C18 100 Å, 5 mm × 300 µm i.d., 160454, Thermo Scientific) in a mobile phase of 2% can with 0.1% formic acid at 10 µL/min. After 3 min of loading, the trap column was switched in-line to EASY spray columns with 50 cm and 75 μm inner diameters (ES803, PepMap RSLC, C18, 2 μm, Thermo Scientific) at 300 nL/min. Separation was performed by mixing A: 0.1% FA and B: 80% CAN with the following gradient: 5 min (2.5% B to 10% B), 120 min (10% B to 30% B), 20 min (30% B to 50% B), 5 min (50% B to 99% B), and 10 min (hold 99% B). The column was then equilibrated with 2.5% B for 17 min. Data acquisition was controlled using Xcalibur 4.0 and Tune 2.9 Software (Thermo Scientific). The mass spectrometer was operated in data-dependent (dd) positive acquisition mode, alternating between a full scan (*m*/*z* 380–1580) and subsequent HCD MS/MS of the 10 most intense peaks from the full scan (normalized collision energy of 27%). The ESI spray voltage was 1.9 kV.

Data analysis was performed with MaxQuant (version 1.6.9.0) using the Andromeda search engine with default search settings, including a 1% false discovery rate at the PSM, peptide, and protein levels. Spectra were aligned with human protein sequences in the UniProt database (data retrieved in January 2023), which contains 47,942 sequences. Mass tolerances for precursor and fragment ions were set at 4.5 and 20 ppm, respectively, during the main search. Enzyme specificity was set at the C-terminal to arginine and lysine, with cleavage at proline bonds also allowing a maximum of two missed cleavages. Variable modifications considered included the oxidation of methionine residues and acetylation of N-terminal protein residues, while the carbamidomethylation of cysteine residues was set as a fixed modification. Matching between runs was activated with a matching time window of 0.7 min and an alignment time window of 20 min. Reverse database hits and proteins identified with a Q-value less than 0.01 were removed, and samples were grouped. Proteins with fewer than three valid values in at least one group were removed, and missing values were imputed from a normal distribution around the detection limit. Protein intensity-based absolute quantification (iBAQ) values from the two programs were preprocessed using three methods: (i) the removal of frequent MS impurities followed by a Log2 (x + 1) transformation; (ii) the removal of frequent MS contaminants followed by a Log2 (x + 1) transformation and quantile normalization; and (iii) the removal of frequent MS contaminants followed by a Log2 (x + 1) transformation, quantile normalization, and abundance filtering to optimize the overall Gaussian distribution of quantitative values. The distribution of quantitative values in each sample was analyzed for each processing method. Statistical analysis was performed using Perseus Software (version 1.6.2.1) to determine the differentially expressed proteins, and the TEST group was compared with the CTRL group.

Functional enrichment was achieved by extracting all functional categories for which at least a sample showed significant enrichment based on the hypergeometric probability test. For these categories, the number of proteins matching the functional category was extracted, and these were processed by multiple corrections of *p*-values (0.05).

### 4.6. Western Blot Analysis

Salivary gland proteins were separated by 12.5% SDS-PAGE, and then, transferred into a nitrocellulose membrane (Amersham™, Protan^®^, GE Healthcare, Buckinghamshire, UK, 0.45 μm porosity) in transfer buffer (25 mM Tris, 192 mM glycine, and 20% methanol) for 2 h at 200 mA. Protein loading was controlled by Ponceau S staining. Nonspecific binding was blocked with 5% (*w*/*v*) nonfat dry milk in tris-buffered saline (TBS) with Tween 20 (TBS-T). Afterwards, each membrane was incubated with the primary antibody (rabbit polyclonal anti-androgen receptor (ARs; 06-680) or rabbit polyclonal anti-estrogen receptor alpha (ERα; 07-662), from Millipore) diluted in a ratio of 1:1000 in blocking solution. The membranes were then washed with TBS-T, incubated with anti-rabbit secondary horseradish peroxidase-conjugated antibody (NA934; GE Healthcare, UK) diluted in a ratio of 1:1000 in blocking solution, and washed again. Detection was carried out with enhanced chemiluminescence (ECL) reagents (WesternBright™ ECL, advansta, California, USA). Images were acquired using the Gel Doc XR system (Bio-Rad^®^, Hercules, CA, USA), and densiometric analysis was performed with Image Lab Software (Bio-Rad^®^, Hercules, CA, USA, version 6.0.0), and values are expressed in arbitrary units.

### 4.7. Gelatin Zymography Analysis

Zymography assay was performed according to Vitorino et al. [74]. Briefly, volumes of submandibular gland extracts equivalent to 50 μg of protein were diluted (1:2) on charging buffer (0.5 M Tris-HCl pH 6.8, 50% (*w*/*v*) SDS, 20% (*v*/*v*) glycerol, and 0.1% (*w*/*v*) bromophenol blue) for 10 min, at room temperature. Then, these samples were loaded in 10% SDS-PAGE separation gel with 0.1% (*w*/*v*) gelatin, and the gels were run at 125V in running buffer (100 mM Tris, 100 mM Bicine, and 0.1% (*w*/*v*) SDS). Next, the gels were incubated in renaturation buffer (2.5% Triton X-100) for 45 min at room temperature and with agitation. Afterwards, one gel was incubated in development buffer (50 mM Tris-HCl pH 8.8, 5 mM NaCl, 10 mM CaCl2∙H2O, 1 μM ZnCl2, pH 7.4, 0.02% (*v*/*v*) Triton X-100), and the other gel with the same samples was incubated in development buffer containing ethylenediamine tetraacetic acid (EDTA; 50 mM Tris-HCl pH 8.8, 10 mM EDTA∙2NA∙2H2O, 5 mM NaCl, pH 7.4, 0.02% (*v*/*v*) Triton X-100) overnight, at 37 °C. The zymography gels were then stained with 0.4% (*w*/*v*) Coomassie blue G250 prepared in 50% ethanol and 10% acetic acid, for 3 h, at room temperature, and with agitation. Lastly, the gels were de-stained with a solution of 25% ethanol and 5% acetic acid. The gels were scanned using a Gel Doc XR system (Bio-Rad^®^, Hercules, CA, USA) and later analyzed with Image Lab Software (Bio-Rad^®^, Hercules, CA, USA, version 6.0.0), in which the optical densities obtained were expressed in arbitrary units.

### 4.8. Statistical Analysis

All values are presented as mean ± standard deviation (SD) of biological replicates of each group. The statistical significance of the differences between the experimental groups was determined using the unpaired Student *t*-test in GraphPad Prism^®^ Software for Windows (version 8.0.1). The level of significance was set at 5%.

## Figures and Tables

**Figure 1 ijms-25-00550-f001:**
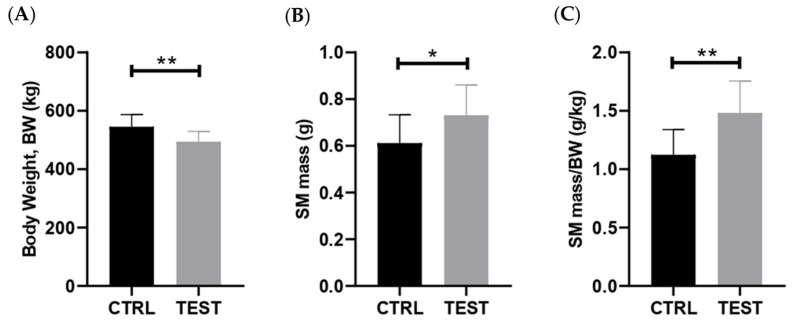
Impact of chronic testosterone exposure on anthropometric parameters in rats at 61 weeks of age (n = 10 for CTRL and n = 14 for TEST). The parameters analyzed were (**A**) body weight; (**B**) submandibular gland (SM) mass; and (**C**) submandibular gland (SM) mass-to-body weight (BW) ratio (* *p* < 0.05; ** *p* < 0.01).

**Figure 2 ijms-25-00550-f002:**
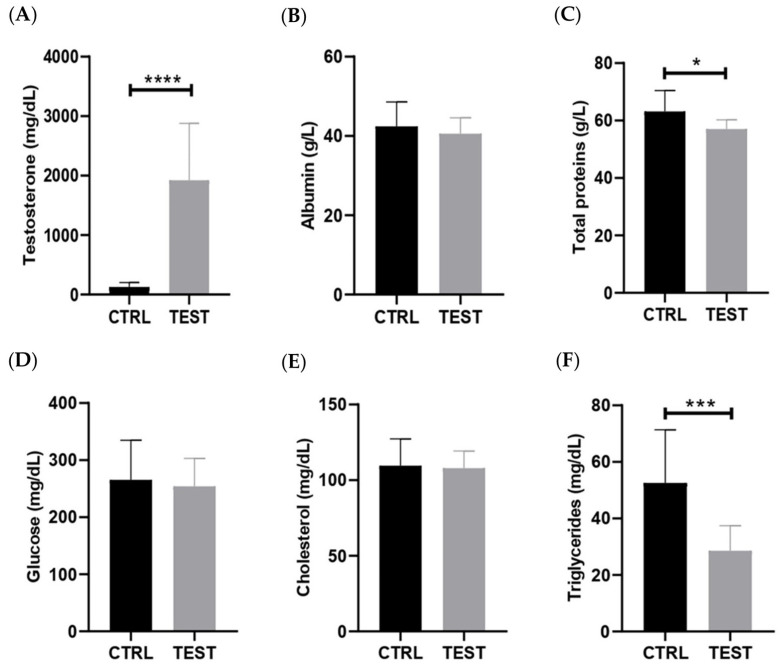
Effect of testosterone exposure on systemic parameters in rats at 61 weeks of age (n = 8 for CTRL and n = 14 for TEST). The parameters analyzed were (**A**) testosterone; (**B**) albumin; (**C**) total proteins; (**D**) glucose; (**E**) cholesterol; and (**F**) triglycerides levels (* *p* < 0.05; *** *p* < 0.001; **** *p* < 0.0001).

**Figure 3 ijms-25-00550-f003:**
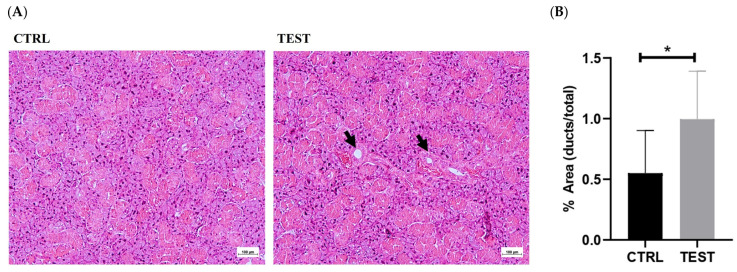
Histological comparison of submandibular gland sections between CTRL and TEST rats. Hematoxylin and eosin (H&E) staining was used to visualize the cellular and tissue structures (**A**). The glandular tissue appears organized, with well-defined secretory acini and ductal structures (intercalated, granular and striated ducts). The acini exhibit typical morphology, with centrally located nuclei and basophilic cytoplasm. Ducts (black arrows) appear lined with cuboidal or columnar epithelial cells. The submandibular glands from the TEST group exhibit larger granular ducts that are lined with a taller epithelium. The ductal-to-whole gland area ratio was also evaluated in both groups (**B**) (* *p* < 0.05).

**Figure 4 ijms-25-00550-f004:**
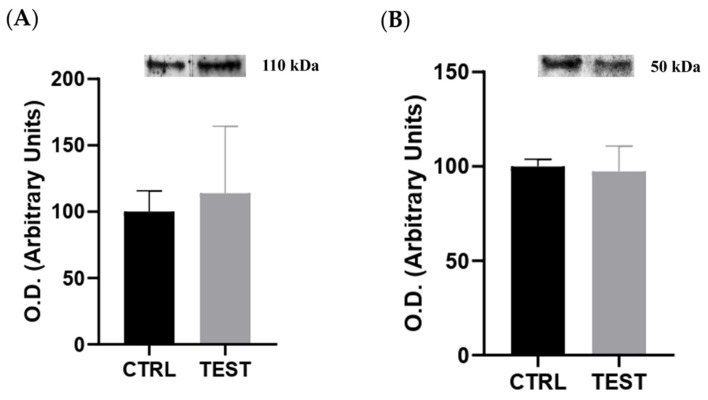
Impact of testosterone administration on the content of (**A**) androgen receptor (n = 5 for CTRL and n = 6 for TEST) and (**B**) estrogen receptor alpha (n = 6 for CTRL and TEST) evaluated through Western blotting. Above each graph are representative immunoblot images. The optical densities (O.D.) obtained are expressed in arbitrary units.

**Figure 5 ijms-25-00550-f005:**
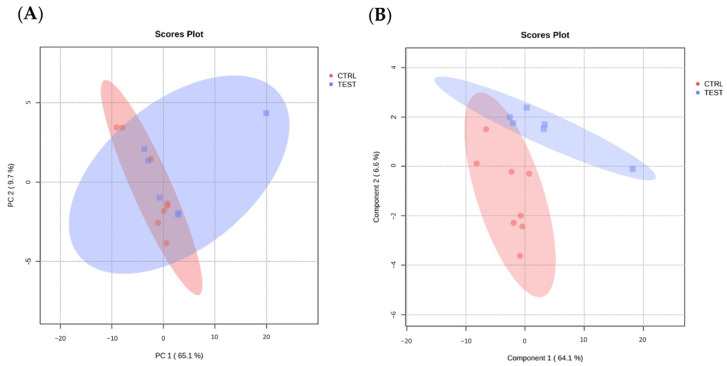
Modeling data projection to optimize separation between CTRL and TEST groups of submandibular glands samples. (**A**) Principal component analysis (PCA) and (**B**) partial least squares discriminant analysis (PLS-DA) of submandibular gland proteome data from animal groups (n = 6 per group), performed with MetaboAnalyst.

**Figure 6 ijms-25-00550-f006:**
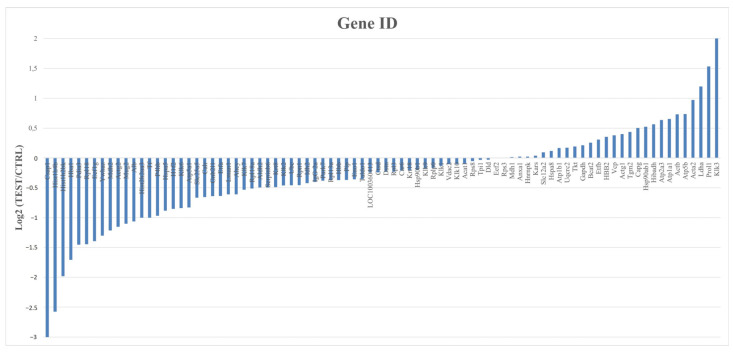
Quantitative analysis of proteins from CTRL and TEST groups with gene identification and corresponding expression levels (Log2). This analysis encompasses all 87 proteins common in both studied groups that appear at least in 4 animals from each group. Proteins were considered with a cut-off of 1.3 for the TEST/CTRL ratio.

**Figure 7 ijms-25-00550-f007:**
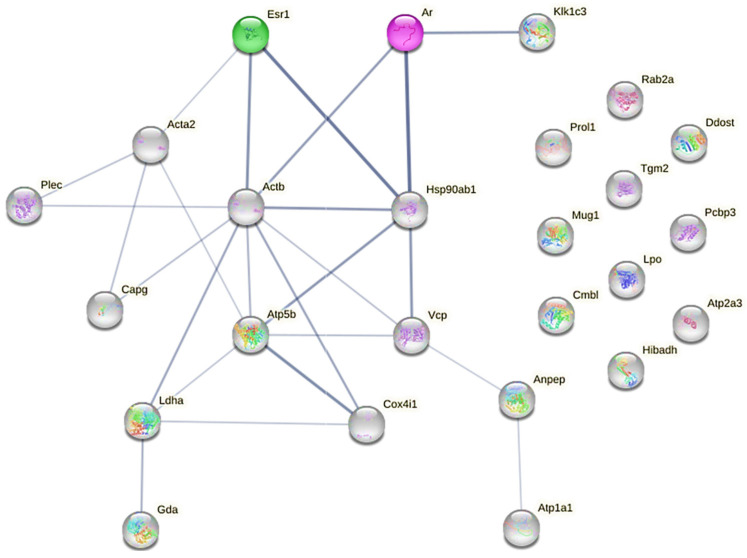
PPI network showing uniquely expressed proteins in TEST group, and common but upregulated ones in TEST group, and sex hormone receptors (AR and ERα). Darker network edges represent a higher level of confidence. Confidence level value range: [0.150; 0.900]. PPI network was constructed with STRING server.

**Table 1 ijms-25-00550-t001:** List of kallikreins identified by SDS-PAGE-LC-MS/MS that potentially contributed to the proteolytic activity in submandibular glands and seen in zymography gel. Fold change TEST/CTRL values are presented.

Gene Name	Protein Name	UniProt Accession	Molecular Weight (Da)	Fold Change (TEST vs. CTRL)
Klk1	Kallikrein-1	P00758	28.852	(=) 0.9114
Klk2	Kallikrein-2	P20151	28.671	(=) 0.7263
Klk3	Glandular kallikrein-3, submandibular	P07288	28.741	(+) 4.2170
Klk6	Prostatic glandular kallikrein-6	P36374	29.013	(−) 0.5583
Klk7	Kallikrein-7	P36373	28.972	(−) 0.6909
Klk9	Submandibular glandular kallikrein-9	P07647	28.368	(=) 0.8797
Klk10	Glandular kallikrein-10	P36375	28.981	(=) 0.9312

Legend: (−) represents downregulated proteins (with fold change TEST/CTRL < 0.7); (=) represents proteins with no considerable expression (fold change TEST/CTRL [0.7–1.3]); (+) represents upregulated proteins (fold change TEST/CTRL > 1.3).

## Data Availability

The data are contained within the article or Appendix A.

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
