# Peer review of "Long-Term Exposure to Supraphysiological Levels of Testosterone Impacts Rat Submandibular Gland Proteome"

_ijms, 2023, doi:10.3390/ijms25010550_

Round 1

Reviewer 1 Report

Comments and Suggestions for Authors

This manuscript describes the effects of supra-physiological doses of testosterone on submandibular glands in male rats treated for 47 weeks. The results show significant changes increase in the submandibular gland and mass significant changes in specific protein levels. The AR and ER were not affected. The results show that exposure to high doses of testosterone over a long period may result in changes in the function and histology of submandibular glands. However, the physiological significance of these effects remains questionable. The study failed to show the possible effects of physiological doses of testosterone on the histology and protein expression in submandibular gland.

1.       The study should include a group of rats with physiological doses of testosterone and castrated male rats to show specificity of these effect of androgens and/or estrogens after aromatization.

2.       The material and methods for treatment of rats do not belong to the current study. The treatment protocol mentions carcinogens and flutamide treatment in addition of testosterone and study effects on prostate.

Comments on the Quality of English Language

No concerns on the quality of English language.

Author Response

Author's Reply to the Reviewer 1

This manuscript describes the effects of supra-physiological doses of testosterone on submandibular glands in male rats treated for 47 weeks. The results show significant changes increase in the submandibular gland and mass significant changes in specific protein levels. The AR and ER were not affected. The results show that exposure to high doses of testosterone over a long period may result in changes in the function and histology of submandibular glands. However, the physiological significance of these effects remains questionable. The study failed to show the possible effects of physiological doses of testosterone on the histology and protein expression in submandibular gland.

  1. The study should include a group of rats with physiological doses of testosterone and castrated male rats to show specificity of these effect of androgens and/or estrogens after aromatization.

R: We acknowledge reviewer’s point of view regarding the current gaps in understanding the intricate mechanisms governing testosterone’s action in the salivary glands and its physiological implications. Nevertheless, we believe that our study adds valuable insights into the impact of this hormone on the remodelling of these glands and potential systemic consequences.

To enhance the comprehensiveness of our investigation, we recognize the merit of including animal groups subjected not only to supraphysiological testosterone levels but also to underphysiological levels, with comparisons to a physiological control group. Such an approach could yield additional information on the nuanced effects of testosterone on these glands. However, in adherence to ethical considerations and the principles of the 3Rs policy to reduce, refine, and replace animal usage, we focused our study on assessing the long-term effects of supraphysiological testosterone levels. This was accomplished by evaluating histological and molecular remodeling of submandibular glands in response to testosterone.

In addressing the concern of underphysiological levels of testosterone, recent studies, such as the work by Han et al. 2022 (https://www.ncbi.nlm.nih.gov/pmc/articles/PMC8991617) have explored the molecular effects of low testosterone levels. This study, involving castrated rats, demonstrated severe degeneration of submandibular glands. Interestingly, this degeneration was not associated with altered levels of AR but rather to changes in downstream targets. Comparison to our results were included in the revised version of our manuscript.

  1. The material and methods for treatment of rats do not belong to the current study. The treatment protocol mentions carcinogens and flutamide treatment in addition of testosterone and study effects on prostate.

R: In fact, the animal protocol implemented was designed to investigate organs adaptations to supraphysiological concentrations of testosterone, namely in prostate remodelling. Given that testosterone is recognized as a carcinogen and tumor-promoting agent for the prostate gland, concerns about the potential increased risk of prostate cancer associated with testosterone therapy have been raised. Its effect may be modulated in prostate by MNU following flutamide sensitization (https://pubmed.ncbi.nlm.nih.gov/25247471). The effect of MNU and flutamide is highly specific for prostate gland in male rats and the pharmacokinetics of flutamide indicate a half-life of just a few hours, and according to models proposed for rats and humans, it practically disappears from tissues after 24 hours (https://pubmed.ncbi.nlm.nih.gov/31841869/). To ensure the thorough evaluation of potential side-effects, macroscopic examinations at necropsy, subsequent histological analysis of tissues and biochemical serum analysis were conducted and revealed no signs of lesions and inflammation, confirming the safety and absence side effects associated with the experimental protocol. It is noteworthy that this protocol has been previously employed to study the impact of supraphysiological levels of testosterone on bone microstructure. In the present study, we extend the application of this protocol to investigate its effects on salivary gland. Both submandibular and parotid glands were collected at necropsy, weighed, and histologically analyzed. However, no indications of testosterone-induced remodeling of parotid gland were observed. To enhance clarity and coherence, amendments have been made in the Material and Methods and Results sections of the revised manuscript to better integrate and convey this important contextual information.

Reviewer 2 Report

Comments and Suggestions for Authors

hello

thank you very much for an interesting paper

please structure more the abstract- highlight the introduction, material, methods, and results, plus briefly describe the paper's aim

the introduction is well written - add at the end of it the main aim of the paper - the last line should be the aim

next chapter please add material and methods and later results

material and methods - briefly describe inclusion and exlusion critiera

create a chart flow if some subjects had been excluded from main study

at the end of discussion add limitations of the study

results are good-please add to the results why this is important in a clinical point of view?

if animals were used - describe briefly the bioethics committee approval and how the animals had been selected for the study and if all had similar background to make this study reliable

the methods section is very nicely written

were there any special results from western-blot testing?

please highlight the top 5 key results and please describe their clinical value

is there any correlation between salivary glands, gathered results and occurrence of some mandibular gland tumors or pathologies?

is decreased/increased salivary flow, or sialolithiasis or other gland diseases taken into consideration while performing the study?

would it be necessary or worth studying if the glands were similar in weight/shape, size in accordance to such studied condition as in the study?

thank you for a great paper

Author Response

Author's Reply to the Reviewer 2

Hello thank you very much for an interesting paper please structure more the abstract- highlight the introduction, material, methods, and results, plus briefly describe the paper's aim the introduction is well written - add at the end of it the main aim of the paper - the last line should be the aim

R: We appreciate your positive feedback. The manuscript adheres to the IJMS journal's guidelines for paper organization. However, in the revised version, certain adjustments were made to the abstract to enhance clarity in the presentation of material and methods, and results, all while staying within the specified word limits.

Next chapter please add material and methods and later results. Material and methods - briefly describe inclusion and exclusion criteria. Create a chart flow if some subjects had been excluded from main study.

R: In accordance with the IJMS guidelines, the material and methods section was appropriately positioned following the discussion section, as per the specified order. The study involved two distinct animal groups: one exposed to testosterone and another not exposed to this hormone. Allocation to each group was determined based on this criterion. Regrettably, one animal in the TEST group expired during the protocol and was subsequently excluded from the analysis. It's important to note that no other animals in either group exhibited signs of developing conditions that could compromise their well-being or introduce bias to the results. Consequently, there was no necessity to exclude additional animals from either group.

To enhance the clarity of the results and following your suggestion, a new supplementary figure has been incorporated (now Figure S1). However, to avoid overwhelming the manuscript with figures and to maintain the focus on the main findings of our study, we have refrained from including an additional figure for the animal protocol.

At the end of discussion add limitations of the study.

R: Following reviewer’s suggestion, we better highlighted the limitations of the present study in the Discussion section of the revised version of the manuscript. Now it can be read: “However, one limitation is that our analysis focused on whole tissue extracts rather than specific subcellular compartments, particularly the nucleus, where the complex hormone/hormone receptor regulates gene expression. Thus, future studies should employ experimental approaches that allow for a more targeted examination of testosterone-AR activity. Further exploration of sex hormone signaling and eventual resistance mechanisms is warranted in future studies to gain a deeper understanding of its contribution to submandibular gland remodeling and, consequently, their impact on overall glandular health outcomes.” In our opinion, this is the main limitation to the understanding of testosterone signaling and, thus, submandibular remodeling in response to this hormone.

Results are good-please add to the results why this is important in a clinical point of view?

R: Following reviewer’s comment, the translational potential of paper findings was clarified at the end of the discussion section in the revised version of the manuscript. The following sentence was added: “These insights hold promise for informing clinical strategies and interventions aimed at optimizing oral health in individuals undergoing testosterone-based therapies.”

If animals were used - describe briefly the bioethics committee approval and how the animals had been selected for the study and if all had similar background to make this study reliable

R: In accordance with IJMS guidelines, the approval of the animal protocol was referenced in the Institutional Review Board Statement, wherein it states: “The animal protocol study was approved by the University of Trás-os-Montes and Alto Douro Ethics Review Body ORBEA (Orgão Responsável pelo Bem-Estar e Ética Animal, reference 424-e-DCV-2016) and by the Portuguese Competent Authority Direção Geral de Alimentação e Veterinária (license number 021326), according to European Guidelines, and following the Portuguese law (decree-law number 113/2013) on animal protection for scientific purposes.” Nevertheless, in response to the reviewer's comment, a clarification was added in section 4.1 of the revised version of the manuscript, explicitly stating: “All the animal experiments were approved by the Institutional Animals Ethics Committee and by Portuguese national authorities (Direção Geral de Alimentação e Veterinária, approval number 021326).”

The methods section is very nicely written, were there any special results from western-blot testing?

R: The western blotting protocol for AR and ER analysis has been routinely executed in our laboratory, including predefined antibody dilutions, protein amount loaded, and incubation times. As such, there are no noteworthy deviations or specific testing results to highlight in this regard.

Please highlight the top 5 key results and please describe their clinical value.

R: As previously indicated, we have enhanced the discussion section to provide a clearer elucidation of our primary findings and underscore the potential translational significance of our results for clinical applications. We believe that the modulation of Klk3, Klk6, and Klk7 expression by testosterone signaling, coupled with their association with the enlargement of submandibular gland ducts, constitutes a novel aspect deserving further exploration, particularly in human tissue, and saliva samples, from individuals undergoing testosterone treatment. Our findings introduce new scientific questions that hold clear clinical implications, paving the way for future research in this domain.

Is there any correlation between salivary glands, gathered results and occurrence of some mandibular gland tumors or pathologies?

R: Addressing the reviewer's concern, we conducted Pearson correlation analyses for the parameters under examination. The statistically significant results have been incorporated into a new figure (now Figure S1) and thoroughly discussed in the document. Notably, our findings revealed a robust positive correlation between testosterone levels and submandibular body weight in rats. Furthermore, a significantly negative correlation was observed between testosterone serum levels and the expression of tissue Klk7.

Is decreased/increased salivary flow, or sialolithiasis or other gland diseases taken into consideration while performing the study?

R: While measuring salivary flow and composition would undeniably provide additional support for our findings, certain technical constraints associated with collecting rats' saliva and the potential stress it might induce on the animals limited its implementation in our study. Importantly, there is no reported relationship between saliva flow rate and testosterone levels, as demonstrated by previous research (Durdiaková et al. 10.1016/j.steroids.2013.09.002., PMID: 24051109).

Additionally, our histological examination of submandibular gland sections did not reveal any indications of sialolithiasis. Throughout the protocol, the rats maintained consistent water and food consumption, and no signs of animal discomfort were observed that might suggest the presence of any disease, including gland-related conditions.

Would it be necessary or worth studying if the glands were similar in weight/shape, size in accordance to such studied condition as in the study?

R: If we understand reviewer’s concern, the variation in submandibular gland-to-body weight aligns with testosterone levels, as evidenced by the positive correlation observed (Figure S1A). The histological features of this gland also exhibit responsiveness to testosterone, although the analysis conducted was more qualitative in nature. We have taken care to better elucidate these associations in the revised version of the manuscript.

Thank you for a great paper.

R: We are grateful for your positive feedback.

Reviewer 3 Report

Comments and Suggestions for Authors

This article makes a significant impact on scientific literature, thanks to its meticulous methodology, enlightening discoveries, and the potential insights it offers into the influence of testosterone on rat submandibular glands. The study prompts additional investigation into the complex relationship between hormonal levels and proteomic alterations, paving the way for future research that could enhance our comprehension of the extensive physiological repercussions associated with hormonal imbalances. 

The article suggests including several additional pieces of information, such as:

-          Whether changes in the size of the salivary gland after testosterone exposure impact the amount of produced saliva.

-          In the discussion, it would be advisable to expand on the topic of kallikrein expression in tumors of salivary glands. What histological subtypes of salivary gland tumors were examined for kallikreins, and what is their clinical relevance in oncological patients?

-          The article lacks a description of potential applications of the study results in a clinical context.

-          In Figure 3, there is a lack of information regarding the objective magnification.

Author Response

This article makes a significant impact on scientific literature, thanks to its meticulous methodology, enlightening discoveries, and the potential insights it offers into the influence of testosterone on rat submandibular glands. The study prompts additional investigation into the complex relationship between hormonal levels and proteomic alterations, paving the way for future research that could enhance our comprehension of the extensive physiological repercussions associated with hormonal imbalances.

The article suggests including several additional pieces of information, such as:

- Whether changes in the size of the salivary gland after testosterone exposure impact the amount of produced saliva.

R: We value the reviewer's insightful comment, which allows us to provide further clarification on this matter. The variation in submandibular gland-to-body weight, as depicted in Figure S1, is indeed influenced by testosterone levels. This phenomenon is attributed, at least in part, to the hypertrophy of granular ducts, as evident in Figure 3. These findings underscore the profound impact of testosterone on the remodeling and, potentially, the function of this gland.

However, it is noteworthy that no discernible differences were found for those proteins involved in regulating vesicle production and transport, among the studied groups. As mentioned earlier, existing literature suggests no direct association between testosterone and salivary production. Nevertheless, our data highlight proteome alterations in Klks, which could have implications for shaping saliva composition. This intriguing aspect poses a scientific question that our results bring to the forefront, paving the way for future studies to delve deeper into the intricate interplay between testosterone and salivary proteome dynamics.

- In the discussion, it would be advisable to expand on the topic of kallikrein expression in tumors of salivary glands. What histological subtypes of salivary gland tumors were examined for kallikreins, and what is their clinical relevance in oncological patients?

R: We appreciate the valuable suggestion, and as a result, in the discussion section of the revised version of the manuscript we have expanded the discussion to incorporate relevant information from the literature concerning changes in the levels of Klks, particularly Klk3, Klk6 and Klk7 that we found modulated by testosterone, and their association with cancer. Regarding tumors, although relatively rare in the salivary glands and inadequately studied, epidemiological studies in humans suggest some sexual dimorphism, indicating a higher incidence of benign tumors in women but not of malignant tumors. While this aspect warrants consideration in future studies, we acknowledge that it may fall beyond the scope of our current work. Our findings hold potential implications for oral health and adds a novel dimension to the understanding of testosterone's influence on submandibular gland dynamics.

- The article lacks a description of potential applications of the study results in a clinical context.

R: In response to concerns raised by reviewers #2 and #3, we have enhanced the discussion to provide a clearer elucidation of the main findings from our study and to underscore the potential translational value of our findings for clinical applications.

- In Figure 3, there is a lack of information regarding the objective magnification.

R: The figure includes a scale bar indicating the magnification.

Round 2

Reviewer 1 Report

Comments and Suggestions for Authors

The authors have not responded to my previous in a way that satisfies the concerns raised on the original submission.

Author Response

We sincerely appreciate Reviewer’s time and effort in reviewing our manuscript entitled “Long-Term Exposure to Supraphysiological Levels of Testosterone Impacts Rat Submandibular Gland Proteome". We are committed to improving the clarity and comprehensiveness of our work.

Response to Reviewer

The authors have not responded to my previous in a way that satisfies the concerns raised on the original submission.

R: We apologize for not addressing the Reviewer's concerns adequately. In reference to the initial concern raised, we are unable to fulfill your suggestion to include a group of rats with physiological doses of testosterone and castrated male rats in our study to demonstrate the specificity of the effects of androgens and/or estrogens after aromatization. This would necessitate a novel animal protocol for which we currently lack ethical permission. Moreover, there are some, still limited, studies conducted with castrated male rats, specifically focusing on salivary gland remodeling (e.g. Chao and Margolius 1983 doi: 10.1210/endo-113-6-2221; Dos Santos et al. 2022 doi: 10.1016/j.jsbmb.2021.106048; Han et al 2022 doi: 10.1186/s12864-022-08521-9). However, the methodological approaches used in those studies differ from the one employed in our research, and they do not encompass the remodeling of the proteome. Despite these differences, we believe that certain relevant comparisons, such as AR and kallikreins content, can be made. We are open to expanding the discussion in our manuscript to address these comparisons more thoroughly. If the Reviewer believes that a more in-depth analysis and discussion of the existing literature on castrated male rats and salivary gland remodeling would strengthen our manuscript, we are committed to incorporating such enhancements.

Regarding the raised concern about the material and methods for rat treatment in our study, particularly in relation to the utilization of carcinogens, flutamide, and testosterone to examine their effects on the prostate, we wish to provide clarification. The protocol in question was implemented to explore the remodeling of various organs in response to testosterone, also focusing on prostate remodeling. However, to address potential misinterpretation, we have taken the Reviewer's feedback into consideration and removed the sentence "Animals from TEST group were submitted to a protocol for induction of prostate cancer" from Section 4.1 of the Material and Methods. Noteworthy is the fact that this animal protocol spanned 49 weeks, equivalent to approximately 25 human years, and the chemicals employed at the outset did not induce lesions in the salivary glands of animals from the TEST group. If the reviewer believes that a more detailed explanation of the protocol is warranted to alleviate any concerns, we are willing to expand upon it.